# Activation and Functions of Col6a1+ Fibroblasts in Colitis-Associated Cancer

**DOI:** 10.3390/ijms25010148

**Published:** 2023-12-21

**Authors:** Niki Chalkidi, Maria-Theodora Melissari, Ana Henriques, Athanasia Stavropoulou, George Kollias, Vasiliki Koliaraki

**Affiliations:** 1Institute for Fundamental Biomedical Research, Biomedical Sciences Research Centre (BSRC) “Alexander Fleming”, 16672 Vari, Greece; 2Institute for Bioinnovation, Biomedical Sciences Research Centre (BSRC) “Alexander Fleming”, 16672 Vari, Greece; 3Department of Physiology, Medical School, National and Kapodistrian University of Athens, 11527 Athens, Greece

**Keywords:** intestinal fibroblasts, heterogeneity, colon cancer

## Abstract

Cancer-associated fibroblasts (CAFs) comprise a group of heterogeneous subpopulations with distinct identities indicative of their diverse origins, activation patterns, and pro-tumorigenic functions. CAFs originate mainly from resident fibroblasts, which are activated upon different stimuli, including growth factors and inflammatory mediators, but the extent to which they also maintain some of their homeostatic properties, at least at the earlier stages of carcinogenesis, is not clear. In response to cytokines, such as interleukin 1 (IL-1) and tumor necrosis factor (TNF), as well as microbial products, CAFs acquire an immunoregulatory phenotype, but its specificity and pathophysiological significance in individual CAF subsets is yet to be determined. In this study, we analyzed the properties of Col6a1-positive fibroblasts in colitis-associated cancer. We found that Col6a1+ cells partly maintain their homeostatic features during adenoma development, while their activation is characterized by the acquisition of a distinct proangiogenic signature associated with their initial perivascular location. In vitro and in vivo experiments showed that Col6a1+ cells respond to innate immune stimuli and exert pro-tumorigenic functions. However, Col6a1+-specific inhibition of TNF receptor 1 (TNFR1) or IL-1 receptor (IL-1R) signaling does not significantly affect tumorigenesis, suggesting that activation of other subsets acts in a compensatory way or that multiple immune stimuli are necessary to drive the proinflammatory activation of this subset. In conclusion, our results show that adenoma-associated CAF subsets can partly maintain the properties of homeostatic fibroblasts while they become activated to support tumor growth through distinct and compensatory mechanisms.

## 1. Introduction

The functional significance of fibroblasts in health and disease is now well established [1,2]. In the intestine, fibroblasts provide structural support and actively interact with surrounding cells to maintain tissue integrity, as well as immunological and epithelial homeostasis [3]. Accordingly, in solid tumors, including those in the intestine, cancer-associated fibroblasts (CAFs) play crucial roles within the tumor microenvironment, contributing structurally and functionally to cancer progression through multiple mechanisms, such as the remodeling of the extracellular matrix, immunoregulation, and modulation of cancer cell proliferation, survival, and metastasis [4,5].

Recent advances in single-cell transcriptomics have broadened our understanding of fibroblast heterogeneity [6,7,8]. In the mouse intestine, three main fibroblast subsets have been described with different origins, locations, and functions under physiological conditions. These include CD81^+^ trophocytes, platelet-derived growth factor receptor alpha (PDGFRa)^hi^ telocytes, and PDGFRa^lo^ CD81^−^ interstitial fibroblasts [3,9]. Similar fibroblast populations have also been described in the human intestine [10,11,12,13]. Several studies in human and mouse tumors have shown that CAFs are a heterogeneous group of cells in terms of their origin, markers, and functions. The majority of CAFs originate from the activation of resident fibroblast populations triggered by signals from cancer cells or the surrounding microenvironment [1,4,5]. Different stimuli are associated with the development of distinct CAF subsets with diverse functions [4,14]. The most notable example is the activation of myofibroblastic (myCAF) and immunoregulatory (iCAF) CAFs by transforming growth factor beta (TGFβ) and interleukin 1 beta (IL-1β), respectively [14,15]. These CAF subsets were initially described in pancreatic ductal adenocarcinoma but have since been identified in many other cancer types [16]. In colorectal cancer, single-cell transcriptomic analyses of human tumors have revealed the presence of both normal-like fibroblasts and CAFs, although there is still no consensus on the number, terminology, origin, properties, and functional significance of individual subsets, and the molecular pathways driving their activation [17,18,19,20].

We have previously shown that innate immune activation of intestinal mesenchymal cells (IMCs) can drive their pro-tumorigenic functions in the intestine [21,22]. Notably, using the *Col6a1Cre* strain, we showed that in vivo deletion of inhibitor of nuclear factor kappa-B kinase subunit b (IKKβ), a key nuclear factor kappa-beta (NF-κB) regulator, led to reduced colon inflammation and associated carcinogenesis in mice [22]. However, deletion of IKKβ in a broader population of mesenchymal cells targeted by the *Col1a2CreER* mouse had the opposite effect, suggesting diverse properties of distinct CAF subsets [23,24]. Further analysis indeed showed that the *Col6a1Cre* strain targets a fraction of IMCs in homeostasis and CAFs in intestinal cancer [22,25]. In more detail, *Col6a1Cre* mice target the majority of PDGFRα^hi^ fibroblasts and pericytes in the colon, which actively contribute to intestinal morphogenesis during development, as well as epithelial cell proliferation and differentiation in homeostasis [25]. However, a detailed characterization of Col6a1+ cells in cancer is still missing.

In this study, our aim was to describe the properties of Col6a1+ CAFs in intestinal carcinogenesis and define their pathophysiological significance, focusing on their innate immune activation. To this end, we analyzed the transcriptomic profile of *Col6a1Cre* positive and negative CAFs from Azoxymethane (AOM)/Dextran sodium sulfate (DSS) colonic adenomas, assessed their pathophysiological role in vitro and in vivo, and defined the in vivo role of IL-1 and TNF-mediated activation in this subset.

## 2. Results

### 2.1. CAFs Partly Maintain Fibroblast Identities upon AOM/DSS Intestinal Carcinogenesis

In our previous work, we have shown that the *Col6a1Cre* mouse targets mainly PDGFRa^hi^ fibroblasts and perivascular cells in the intestine [22,25]. To define the role of these cells in intestinal carcinogenesis, we crossed *Col6a1Cre* mice with the TdTomato-to-GFP replacement (mTmG) reporter strain (hereafter *Col6a1*-mTmG) and subjected them to the AOM/DSS model of colitis-associated cancer (CAC) [26]. We isolated colonic adenomas, and after exclusion of immune, endothelial, erythroid, and epithelial cells, using the lineage negative (Lin−) markers CD45, CD31, Ter119, and EpCAM, respectively, we isolated Col6a1+/GFP+ and Col6a1−/GFP− (or Tomato+) CAFs by FACS sorting and performed 3′ mRNA sequencing (Figure 1A,B). Results were compared with previously published data of Col6a1+ and Col6a1− cells in homeostasis [25].

To examine whether the transcriptomic identity of IMCs is maintained in CAC, we assessed the expression of a selected gene signature, previously described for Col6a1+ and Col6a1− cells in homeostasis [25]. This analysis showed that Col6a1+ and, to a lesser extent, Col6a1− CAFs maintain some of their physiological characteristics (Figure 1C). Col6a1+ CAFs, in particular, showed increased expression of telocyte markers (*Foxl1*), BMPs (*Bmp3*, *Bmp5*), and Wnt signaling regulators (*Wnt5a*), which play a significant role in homeostatic epithelial cell differentiation [3,9,25]. Notably, Col6a1+ CAFs expressed CD201, which we have previously identified as a marker of PDGFRa^hi^ fibroblasts and pericytes [25] and localized adjacent to neoplastic cells, similar to their normal counterparts (Figure 1D,E) [25]. Conversely, Col6a1− CAFs sustained their increased expression of stem cell maintenance mediators and trophocyte markers, such as *Wnt2*, *Cd34*, and *Pi16* (Figure 1C) [3,9,25]. It should be noted that this subset includes all trophocytes and the majority of PDGFRα^lo^ fibroblasts, which we cannot distinguish in these experiments. In addition, Col6a1+ CAFs showed enrichment in genes associated with vasoconstriction and blood pressure, which is consistent with the targeting of pericytes by the *Col6a1Cre* mouse (Figure 1C) [25]. To further validate this result, we compared the genes enriched in Col6a1+ CAFs versus Col6a1− CAFs with a recently published mural gene signature [27]. Interestingly, 33 of the 45 genes of this signature were also found in Col6a1+ CAFs, thus supporting the perivascular role of Col6a1+ CAFs in colitis-associated carcinogenesis (Figure 1F). Overall, these results suggest that CAFs in AOM/DSS-induced adenomas originate from resident mesenchymal cells, including different fibroblast subsets and pericytes, and partly maintain their physiological properties.

### 2.2. Col6a1+ and Col6a1− CAFs Are Activated in AOM/DSS Colon Carcinogenesis

Despite the maintenance of a minimum homeostatic gene expression signature in Col6a1+ and Col6a1− CAFs, deregulated gene expression analysis in comparison to homeostatic Col6a1+ and Col6a1− IMCs, respectively, showed significant alterations in the gene expression of both subsets. This analysis showed 1045 upregulated and 540 downregulated genes in the Col6a1+ CAFs and 1906 upregulated and 1320 downregulated genes in the Col6a1− CAFs (Figure 2A,B). Comparison between the deregulated genes in the two subsets revealed both common and distinct transcriptional signatures during cancer (Figure 2C and Appendix A), which was further analyzed through pathway enrichment and network associations using Metascape.org [28] (Figure 2D–F and Appendix A).

The upregulated gene signature shared between Col6a1+ and Col6a1− CAFs (709 genes) was enriched in biological functions related to epithelial cell differentiation, proliferation, and development, indicating that both Col6a1+ and Col6a1− CAFs can directly affect neoplastic cells and drive cancer growth. Accordingly, they can also modulate cell death pathways (formation of the cornified envelope, pyroptosis), whose inhibition is important for cancer progression. Terms associated with cell-cell adhesion and junction organization indicate that CAF activation results in increased cellular interactions, including those between CAFs (muscle-cell adhesion). Notably, several enriched terms (inflammatory response, neutrophil chemotaxis, and response to bacterium) further support the immunoregulatory role of CAFs in CAC and, more specifically, their functions in innate immune responses [1,4,5] (Figure 2D).

Analysis of Col6a1+ CAF’s uniquely upregulated genes revealed significant enrichment in pathways related to blood vessel development, vascular endothelial proliferation, blood circulation, and hemostasis, indicating that Col6a1+ CAFs play a significant role in tumor-associated angiogenesis (Figure 2E). Examples of genes enriched in this process include those encoding for integrins (*Itga7*, *Itga4*, *Itgb3*), molecules of the Notch signaling pathway (*Jag1*, *Hey2*), as well as *Rgs5* and *Angpt4* that are typically expressed by pericytes and smooth muscle cells [29] (Appendix A). Other enriched pathways, specifically in Col6a1+ CAFs, include those involved in their activation, both towards a myofibroblastic (mesenchymal cell development, muscle contraction) and inflammatory phenotype (response to chemokine). Notably, genes associated with GPCR signaling are significantly upregulated, suggesting a role of GPCR in Col6a1+ fibroblast activation (Figure 2E).

Col6a1− CAFs were highly enriched in metabolic pathways, including the metabolism and biosynthesis of lipids (*Acadl*, *Alb*, *Cpt2*, etc.), amino acids (*Arg2*, *Cs*, *Gss,* etc.), organic acids (*Cftr*, *Slc26a3*, *Acsl1*, etc.), and small molecules (*Alox12*, *Cbs*, *Edn2*, etc.). They also expressed genes that are involved in the secretion and transport of small molecules (various solute carrier family members, *Apoc4*, *Heph*, etc.), suggesting increased secretory functions (Figure 2F and Appendix A). Other enriched terms included unique genes related to immune response and epithelial differentiation, further supporting the pro-tumorigenic role of Col6a1− CAFs in CAC (Figure 2F and Appendix A).

These findings show that both Col6a1+ and Col6a1− IMCs cells are activated in AOM/DSS-induced colitis-associated carcinogenesis to exert both similar and unique pro-tumorigenic functions. Similar transcriptional profiles suggest an important role both in epithelial cell proliferation and in immune regulation. Unique transcriptional profiles highlight a significant role for Col6a1+ CAFs in tumor-associated angiogenesis and reveal an increased metabolic and secretory phenotype for Col6a1− CAFs.

### 2.3. Col6a1+ and Col6a1− CAFs Support Cancer Cell Growth In Vitro and In Vivo

To define the physiological importance of the two CAF subsets on cancer growth, we next performed a series of in vitro and in vivo experiments. Initially, we assessed the colony formation potential of Caco-2 colon cancer cells grown on top of either Col6a1+ or Col6a1− IMCs. The size of Caco-2 colonies on Col6a1+ IMCs after 3 days in culture was statistically significantly larger in comparison to Col6a1− IMCs; however, the difference was small (Figure 3A–C). We then used a more physiologically relevant in vitro model by co-culturing Col6a1+ or Col6a1− cells with AOM/DSS tumoroids for 3 days. The size of tumoroids was similar in both co-culture conditions (Figure 3D–F). These results show that both fibroblast subsets can support cancer cell proliferation.

To further examine the in vivo role of the two subsets in cancer growth, we performed allograft experiments. We isolated fresh colonic Col6a1+ and Col6a1− IMCs through FACS sorting and co-injected them with MC38 colon cancer cells subcutaneously in the flanks of C57/Bl6 wild-type mice. MC38 cells alone were used as controls. After 15 days, allograft tumors containing either Col6a1+ or Col6a1− IMCs showed no difference in tumor size between the two subsets (Figure 3G). Notably, both IMC subsets were equally represented inside the tumor allografts, as shown by fluorescent microscopy (Figure 3H). Taken together, our results show that both Col6a1+ and Col6a1− IMCs can support cancer growth in vitro and in vivo.

### 2.4. Deletion of IL-1R1 or TNFR1 in Col6a1+ IMCs Is not Sufficient to Ameliorate Colitis-Associated Carcinogenesis

Previous studies have shown that CAFs respond to cytokines (TNF, IL-1β) in the tumor microenvironment to drive their proinflammatory activation [1,4,14]. We have also shown that innate immune sensing by fibroblasts plays a significant role in intestinal tumorigenesis [21,22]. Fibroblast-specific IKKβ deletion using the *Col6a1Cre* mouse resulted in decreased colitis-associated tumorigenesis due to reduced inflammation [22]. To better understand the response of IMCs to cytokines and innate immune signals, we stimulated unsorted intestinal fibroblasts with TNF, IL-1β, and LPS and analyzed their secretome using a Proteome Profiler Array (Figure 4A,B). We found that IMCs respond to all three factors, and the most robust response was upon LPS and IL-1β stimulation (Figure 4A,B). Overexpressed secreted mediators included mainly cytokines (IL-6, IL-11, IL-23, and IL-1β), chemokines (e.g., CXCL1, CXCL2, and CXCL10) and matrix metalloproteinase (MMPs) (MMP-3, MMP-9) (Figure 4A,B). We then used the CXCL2 (or MIP-2) chemokine as a readout and measured the response of sorted Col6a1+ or Col6a1− IMCs in culture supernatants. Interestingly, both IMC subsets responded similarly to all three inducers in vitro (Figure 4C), suggesting that different intestinal fibroblast subsets can respond to inflammatory stimuli and become activated.

Given our previous results on the pro-inflammatory and pro-tumorigenic role of Col6a1+ cells in CAC, we assessed whether in vivo IL-1β or TNF signaling could be upstream of NF-κB activation. For this reason, we crossed *Il1r1^f/f^* [30] and *p55^f/f^* [31] mice with the *Col6a1Cre* strain to specifically inhibit the IL-1R and TNFR pathways in Col6a1+ IMCs and then subjected the mice to the AOM/DSS protocol of colitis-associated carcinogenesis. Both *Il1r1^IMCko^* and *p55^IMCko^* mice developed an equal number of tumors in comparison with their littermate controls (Figure 4D,F). Colon length and colitis scoring were also similar between control and experimental mice, indicating similar levels of inflammation (Figure 4E,G and Appendix A). Notably, similar results were previously also published for Col6a1-specific *Tlr4* deletion [21]. Collectively, these results show that the deletion of a single inflammatory inducer in Col6a1+ cells is not sufficient to reduce inflammation and tumorigenesis, indicating that potential synergistic activation of NF-κB could be driving the pro-inflammatory properties of these cells. Nevertheless, we cannot exclude that deletion of a single inducer in a larger fibroblast subset or in cells preferentially activated by it due to their microenvironment milieu could have a significant effect.

## 3. Discussion

During the last decade, many studies have highlighted the heterogeneity of CAF’s functional characteristics, which may be attributed to their different origins at the cell type (fibroblasts, epithelial cells, endothelial cells, etc.) or subpopulation level [1,4,5]. In this study, we explored the properties and pathophysiological role of Col6a1+ and Col6a1− CAFs in mouse AOM/DSS-induced colitis-associated carcinogenesis, as we have previously shown that the two subsets represent distinct fibroblast populations in the colon [25]. In more detail, colonic cells targeted by the *Col6a1Cre* strain include PDGFRα^hi^ telocytes, pericytes, and a small fraction of PDGFRα^lo^ fibroblasts, while Col6a1− fibroblasts include trophocytes and the majority of PDGFRα^lo^ fibroblasts [25]. Gene expression profiles of the two subsets in colitis-associated adenomas also showed significant differences between them, supporting their diverse identities in cancer. Interestingly, they also displayed a significant transcriptional similarity with their respective normal fibroblast subsets, indicating that CAFs can maintain some of their key physiological identities at least at the early stages of adenoma development. More specifically, Col6a1+ CAFs displayed a dual gene expression signature, reflecting a role both in epithelial cell differentiation and vascular function, similar to their normal counterparts. Accordingly, Col6a1− CAFs maintained some trophocyte markers (*Pi16*) and stem cell maintenance mediators (*Wnt2b* and *Grem1*). This similarity in the gene expression signatures of CAF and normal fibroblast subsets supports the significance of resident fibroblast populations as a source of intestinal CAFs upon colitis-induced carcinogenesis. It also indicates that activation of CAFs could be a stepwise process towards their complete reprogramming at the late stages of the disease. Our results are in line with already published data from human single-cell transcriptomic studies, which show that CAFs in colorectal cancer include normal-like fibroblasts that can be subdivided into subsets, similar to the normal tissue [18,19,32].

Focusing on the molecular pathways governing the activation of the two CAF subsets in AOM/DSS-induced tumors, we found that both Col6a1+ and Col6a1− CAFs displayed significant deregulation of genes implicated in carcinogenesis in comparison to normal fibroblasts. These included both common and uniquely deregulated expression profiles between the two subsets. Pathway analysis highlighted the significant and unique enrichment of Col6a1^+^ cells in functions related to angiogenesis, which agrees with the targeting of both pericytes and fibroblasts near capillaries by the *Col6a1Cre* mouse in homeostasis [25]. Notably, perivascular CAFs have been detected in multiple human tumors and have been shown to exert pro-tumorigenic properties through the regulation of endothelial cell proliferation and function [16,33,34]. Accordingly, single-cell analyses of human colorectal tumors have revealed an abundant Rgs5-expressing subset, as well as increased expression of pericyte markers, such as *Pdgfrb*, *Mcam*, etc. [17,18,19,20,32]. Col6a1− CAFs, on the other hand, displayed a unique activated gene signature representing an enhanced metabolic and secretory activity, in agreement with the secretory and metabolic reprogramming of CAFs [1,4,5]. Despite these significant differences, Col6a1+ and Col6a1− fibroblasts displayed similar pro-tumorigenic properties in both in vitro co-culture assays with cancer organoids and in vivo allograft experiments. These experiments suggest that despite their differences, similar and complementary activated pathways in both subsets are sufficient to support cancer cell proliferation and growth. Indeed, the two subsets showed increased expression levels of common or functionally complementary genes related to epithelial and cancer cell proliferation, evasion of apoptosis, and immunoregulation.

Functions related to the regulation of immune responses are particularly important in CAC and the AOM/DSS model since they are driven by tissue damage and concomitant inflammation. We have shown in the past that deletion of immune-related genes, such as *Ikk2* and *Myd88* in Col6a1+ fibroblasts, results in reduced spontaneous and/or colitis-associated carcinogenesis [21,22]. Indeed, we show here that intestinal fibroblasts respond to common immune stimuli, such as IL-1β, TNF, and LPS, and induce a robust pro-inflammatory response, including cytokines, chemokines, matrix metalloproteinases, and other inflammatory mediators. Notably, we show that both Col6a1+ and Col6a1− subsets respond similarly to innate stimuli, at least in vitro, and are thus not characterized by inherent differences in the ability to mount an inflammatory response. Deletion of IL-1R1 or TNFR1, specifically in *Col6a1Cre*-expressing cells, was not sufficient to reduce colon tumorigenesis in the AOM/DSS models. Accordingly, deletion of TLR4 signaling in these cells had a similar effect [21]. This may seem to be in contrast with the anti-tumorigenic effect of IKKβ deletion using the same genetic tools and mouse model [22]. However, each one of these stimuli can drive activation of ΝF-κB signaling, and the absence of more than one may be necessary to dampen ΝF-κB in these cells. Furthermore, since Col6a1− cells are also able to respond to inflammatory stimuli, the microenvironmental milieu of individual cells plays a crucial role in their in vivo activation. As such, deletion of individual receptors in another or in multiple subsets may be sufficient to inhibit carcinogenesis in vivo. Indeed, IL-1 signaling in Grem1^+^ cells is sufficient and necessary for recovery after DSS-induced colitis [35].

Overall, our study offers valuable insights into the origins, pathophysiological role, and molecular pathways governing CAF activation in the AOM/DSS model of colitis-associated carcinogenesis, highlighting the complementarity of CAF subset-specific activator signals and functions.

## 4. Materials and Methods

### 4.1. Mice and Study Approval

*Col6a1Cre* [36], *Il1r1^f/f^* [30] (a gift from Emmanuel Pinteaux, Ari Waisman, and Werner Muller), and *p55TNFR1^f/f^* [31] mice have been previously described. *Rosa26-mT/mG* mice were purchased from the Jackson Laboratory [37]. All mice were maintained on a C57/Bl6 background, and experiments were performed using littermate, co-housed control, and experimental mice. Both male and female mice were used at the ages of 2–6 months. Mice were maintained under specific pathogen-free conditions in the Animal House of the Biomedical Sciences Research Center “Alexander Fleming”. Animal studies were approved by the Institutional Committee of Protocol Evaluation in conjunction with the Veterinary Service Management of the Hellenic Republic Prefecture of Attika according to all current European and national legislation under the permissions: 1253-10/03/2014, 5759-26/11/2015, 8443-18/01/2017, 8450-18/01/2017, and 420750-22/06/2020. Experiments were performed in accordance with the guidance of the Institutional Animal Care and Use Committee of BSRC “Alexander Fleming’’.

### 4.2. Induction of Colitis-Associated Cancer

AOM/DSS colitis-associated cancer (CAC) was induced according to previously published protocols [26]. Briefly, 8–10-week-old mice received a single intraperitoneal injection of AOM (Sigma, St. Louis, MO, USA) followed by three cycles of 2.5% DSS (MW: 36,000–50,000 Da, MP Biomedicals, Santa Ana, CA, USA) in the drinking water. Each DSS cycle was followed by 14 days of regular water. Colitis was monitored by measuring weight loss. At the end of the protocol, mice were sacrificed, the colon was resected, and its length was measured as an indicator of colitis severity. The number of macroscopically visible tumors was counted.

### 4.3. FACS Analysis and Sorting

Colonic tissues or colonic tumors were dissected, washed with HBSS (Gibco, Waltham, MA, USA), containing antibiotic antimycotic solution (Gibco, Waltham, MA, USA), and digested. The colonic tissue was digested in 300 U/mL Collagenase XI (Sigma-Aldrich, St. Louis, MO, USA) and 1 mg/mL Dispase II (Roche, Basel, Switzerland) in DMEM for 60 min at 37 °C. The colonic tumors were digested in 1000 U/mL Collagenase IV (Sigma-Aldrich, St. Louis, MO, USA), 1 mg/mL Dispase II (Roche, Basel, Switzerland), and 100 U/mL Dnase I (Sigma-Aldrich, St. Louis, MO, USA) in DMEM (BioSera, France) in three serial 20-min digestions to increase cell viability. The cell suspension was centrifuged, washed three times in FACS buffer (5% FBS (Biowest, United Kingdom) in PBS), and cells were counted. For staining, 1–2 million cells/100 μL were incubated with the antibodies shown in Table 1. Propidium Iodide (Sigma, St. Louis, MO, USA) was used for live-dead cell discrimination. Sample analysis was performed using the FACSCanto II flow cytometer (BD, San Jose, CA, USA) or the FACSAria III cell sorter (BD, San Jose, CA, USA) and the FACSDiva (BD, San Jose, CA, USA) or FlowJo software (v10.2, FlowJo, LLC, Ashland, OR, USA).

Cultured IMCs isolated from *Col6a1^mTmG^* mice were used for sorting at passage 2 based on their GFP or Tomato fluorescent protein expression using a FACSAria III Cell Sorter (BD Biosciences, San Jose, CA, USA). Cells were grouped as Col6a1+ and Col6a1− IMCs for subsequent experiments.

### 4.4. 3′ RNA Sequencing and Analysis

RNA from FACS-sorted cells was isolated using the Single Cell RNA Purification kit (Norgen Biotek, Thorold, ON, Canada). 3′RNA sequencing and analysis were performed as previously described [25]. In more detail, the quantity and quality of RNA samples were analyzed using the Agilent RNA 6000 Nano on an Agilent bioanalyzer. RNA samples with RNA Integrity Number (RIN) > 7 were used for library preparation using the 3′ mRNA-Seq Library Prep Kit Protocol for Ion Torrent (QuantSeq-LEXOGEN™, Vienna, Austria), according to the manufacturer’s instructions. The quantity and quality of libraries were assessed using the DNA High Sensitivity Kit in the bioanalyzer, according to the manufacturer’s instructions (Agilent, Santa Clara, CA, USA). Libraries were pooled and templated using the Ion PI IC200 Chef Kit (ThermoFisher Scientific, Waltham, MA, USA) on an Ion Proton Chef Instrument or Ion One Touch System. Sequencing was performed using the Ion PITM Sequencing 200 V3 Kit and Ion Proton PI™ V2 chips (ThermoFisher Scientific, Waltham, MA, USA) on an Ion Proton™ System, according to the manufacturer’s instructions. The RNA-Seq FASTQ files were mapped using TopHat2 (version 2.1.1) [38], with default settings and using additional transcript annotation data for the mm10 genome from Illumina iGenomes (https://support.illumina.com/sequencing/sequencing_software/igenome.html, accessed on 30 March 2019). According to the Ion Proton manufacturer’s recommendation, the reads, which remained unmapped, were submitted to a second round of mapping using Bowtie2 (version 1.3.1) [39] against the mm10 genome with the very-sensitive switch turned on and merged with the initial mappings. Through the metaseqr R package (version 4.3.2) [40], GenomicRanges and DESeq were employed to summarize the bam files of the previous step to read counts table and to perform differential expression analysis (after removing genes that had zero counts over all the RNA-Seq samples).

Downstream bioinformatics analysis and visualization were performed using InteractiveVenn for Venn diagrams (www.interactivenn.net, accessed on 30 October 2023) [41] and Metascape (metascape.org, accessed on 30 October 2023) for network plots [28]. Heatmaps were generated in R using an in-house developed script utilizing the package pheatmap (version 1.0.12, https://cran.rproject.org/web/packages/pheatmap/index.html, accessed on 30 October 2023) [42]. We used the Log2-transformed normalized counts of genes for each replicate and performed a z-score transformation for each gene across conditions. Finally, the clustering of gene expression was performed based on pheatmap’s default settings. RNA-seq datasets have been deposited in NCBI’s Gene Expression Omnibus [43] and are accessible through the GEO Series accession number GSE247089.

### 4.5. Isolation and Culture of Primary Mouse Intestinal Mesenchymal Cells

IMC isolation was performed as previously described [44]. Briefly, colons from 6- to 10-week-old mice were isolated, flushed, and washed with ice-cold HBSS (Gibco, Waltham, MA, USA). The epithelial layer was removed after treatment with pre-warmed 5 mM EDTA (Acros Organics, Antwerp, Belgium) and 1 mM DTT (Sigma, St. Louis, MO, USA) in HBSS for 20 min at 37 °C. The remaining colonic tissue was then digested using 300 u/mL Collagenase XI (Sigma, St. Louis, MO, USA) and 0.1 mg/mL Dispase II (Roche, Basel, Switzerland) for 60 min at 37 °C. Samples were filtered through a 70 μm cell strainer, centrifuged, and the cell pellet was resuspended in complete culture DMEM medium (BioSera, France) supplemented with 10% FBS (Biowest, Riverside, United Kingdom), 100 U/mL penicillin/100 mg/mL streptomycin (Gibco, Waltham, MA, USA), 2 mM L-Glutamine (Gibco, Waltham, MA, USA), 1 μg/mL amphotericin B (Sigma, St. Louis, MO, USA), and 1% non-essential amino acids (Gibco, Waltham, MA, USA). Cells were plated in culture flasks and passaged 3–4 times.

### 4.6. Tumor Organoids and Co-Culture with IMCs

Tumor organoids were isolated from AOM/DSS-induced tumors and cultured as previously described [45]. Briefly, colonic tumors were isolated, washed, treated with chelation buffer, and digested using 200 U/mL type IV collagenase, 125 μg/mL type II dispase in DMEM) for 2 h at 37 °C. 10.000 tumor fragments were plated at 24-well plates in 30 μL of Matrigel (Cat. No. 356255, Corning, NY, USA). After passaging tumoroids at a 1:4 ratio, they were mixed with 15.000 sorted cultured Col6a1+ or Col6a1− IMCs in 30 μL Matrigel (Corning, NY, USA) in 48 well plates and co-cultured for 72 h. Images were acquired with the Zeiss Axio Observer Z1 microscope. Organoid measurements were performed using the Fiji/ImageJ software (version 1.53).

### 4.7. Caco-2 Co-Culture Assay

The Caco-2 co-culture assay was performed as previously described [46]. In brief, cultured IMCs were sorted based on GFP expression and plated in 48 well plates. After forming a monolayer, 37.500 Caco-2 colon cancer cells were added for 72 h. Images were acquired with the Zeiss Axio Observer Z1 microscope. Colony size was measured using the ImageJ/Fiji software (version 1.53).

### 4.8. Allografts

*Col6a1*-mTmG mice (6 mice/experiment) were sacrificed at a 6–10-week-old age and their colons were dissected and processed, as described in the FACS analysis section. Epithelial, endothelial, immune, and erythroid cells were excluded through negative selection, and cells were sorted based on their GFP/Tomato expression, as Col6a1+ and Col6a1− IMCs. Subcutaneous injections were performed in 6-week-old mice with 10.000 MC38 mouse colon cancer cells mixed with 100.000 Col6a1+ or Col6a1− IMCs in 100 μL DMEM containing 100 U/mL penicillin/100 mg/mL streptomycin (Gibco, Waltham, MA, USA), 2 mM L-Glutamine (Gibco, Waltham, MA, USA), and 1% non-essential amino acids (Gibco, Waltham, MA, USA). 10.000 MC38 cells alone were also injected as controls. Each experimental group included 8 mice. After 15 days, the mice were sacrificed, the allografts were removed, and their width and length were measured using a caliper. Tumor volume was calculated using the ellipsoid volume formula (π/6) × width × length^2^ [47].

### 4.9. Immunohistochemistry

For histopathology, colon tissues were fixed overnight in 10% formalin and embedded in paraffin. 4-μm sections were mounted on slides and stained with hematoxylin and eosin (Sigma-Aldrich, St. Louis, MO, USA). Colitis and inflammation score was assessed as previously described [22].

For immunofluorescence, AOM/DSS-induced colonic tumors were isolated, fixed with 4% PFA/PBS overnight, and serially immersed in sucrose solutions (15% and 30%). Tumors were then embedded in OCT (VWR Chemicals, Radnor, PA, USA), and cryosections were prepared using the LEICA CM1950 cryotome. Staining was performed using the anti-aSMA antibody (1:100, Sigma, St. Louis, MO, USA) and an anti-mouse-A647 secondary antibody (1:500, Invitrogen, Waltham, MA, USA). A mounting medium containing DAPI (Sigma-Aldrich, St. Louis, MO, USA) was used to stain the nuclei. Images were acquired with a Leica TCS SP8X White Light Laser confocal system.

### 4.10. Proteome Profiling

For cytokine determination assays, cells were plated, serum-starved overnight, and stimulated with 2 mg/mL LPS from *E. coli* (Sigma, St. Louis, MO, USA), 10 ng/mL IL-1β (Peprotech, Cranbury, NY, USA), and 10 ng/mL TNF (Sigma, St. Louis, MO, USA) for 24 h. The expression of a variety of secreted mediators was assessed using the Proteome ProfilerTM Array (R&D Systems, Minneapolis, MN, USA Catalog Number: ARY028) according to manufacturer’s instructions. Quantification of signal intensity was performed using the ChemiDoc XRS+ instrument and the Image Lab software (version 5.2, Bio-Rad, Hercules, CA, USA).

### 4.11. MIP-2 Elisa

Cells were plated, serum-starved overnight, and stimulated with 10 ng/mL IL-1β (Peprotech, Cranbury, NY, USA), 10 ng/mL TNF (Sigma, St. Louis, MO, USA), and 2 mg/mL LPS from *E. coli* (Sigma, St. Louis, MO, USA). Supernatants were collected at 24 h, and MIP-2 quantification was performed using the mouse MIP-2 ELISA kit (Peprotech, Cranbury, NY, USA), according to the manufacturer’s instructions.

### 4.12. Statistical Analysis

Data were analyzed using the GraphPad Prism v8 software. Statistical significance was calculated by Student’s *t*-test, and *p* values ≤ 0.05 were considered significant. Data are presented as mean ± SD.

## Figures and Tables

**Figure 1 ijms-25-00148-f001:**
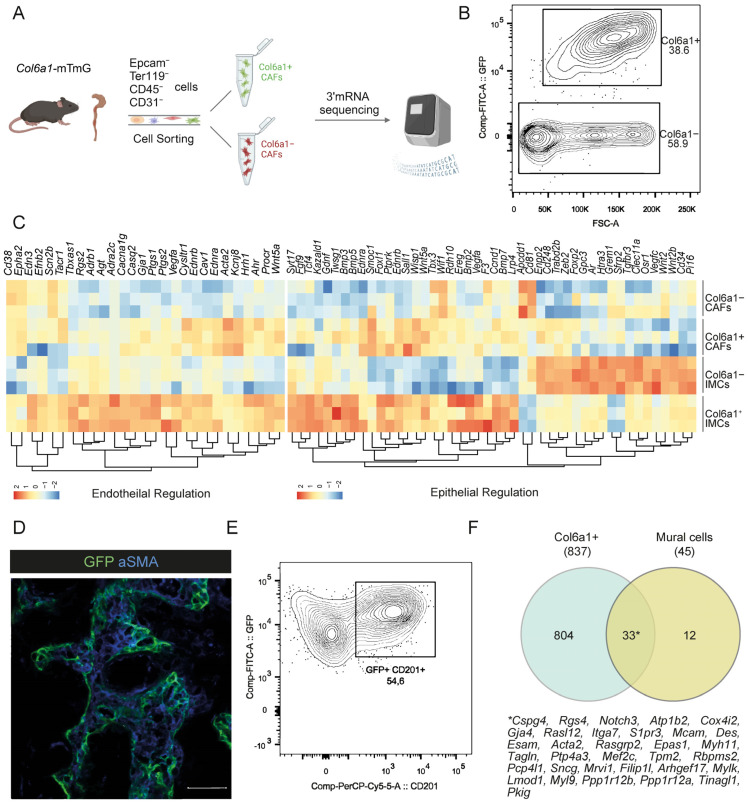
Col6a1+ CAFs and normal fibroblasts display transcriptional similarities. (**A**) Schematic showing the procedure for bulk RNA sequencing of fibroblasts isolated from AOM/DSS-induced adenomas (prepared using Biorender.com). A total of three samples were used for bulk RNA sequencing. Each sample originated from a pool of tumors from five to six mice. (**B**) FACS plot showing the sorting strategy for Col6a1+ and Col6a1− CAFs. (**C**) Heatmap showing gene expression signatures of Col6a1+ and Col6a1− fibroblasts in homeostasis (IMCs) and CAC (CAFs). Log2-transformed normalized read counts of genes for each replicate are shown. Red denotes high expression, and blue denotes low expression values. Read counts are scaled per column. (**D**) Immunohistochemistry for αSMA in AOM/DSS-induced adenomas of *Col6a1*-mTmG mice (n = 5 mice, Scale bar: 50 μm). (**E**) Representative FACS analysis of CD201 expression in Lin−GFP+ cells in AOM/DSS-induced adenomas of *Col6a1*-mTmG mice (n = 2 mice). (**F**) Venn diagram showing similarities in gene expression between Col6a1+ CAFs and mural cells [27].

**Figure 2 ijms-25-00148-f002:**
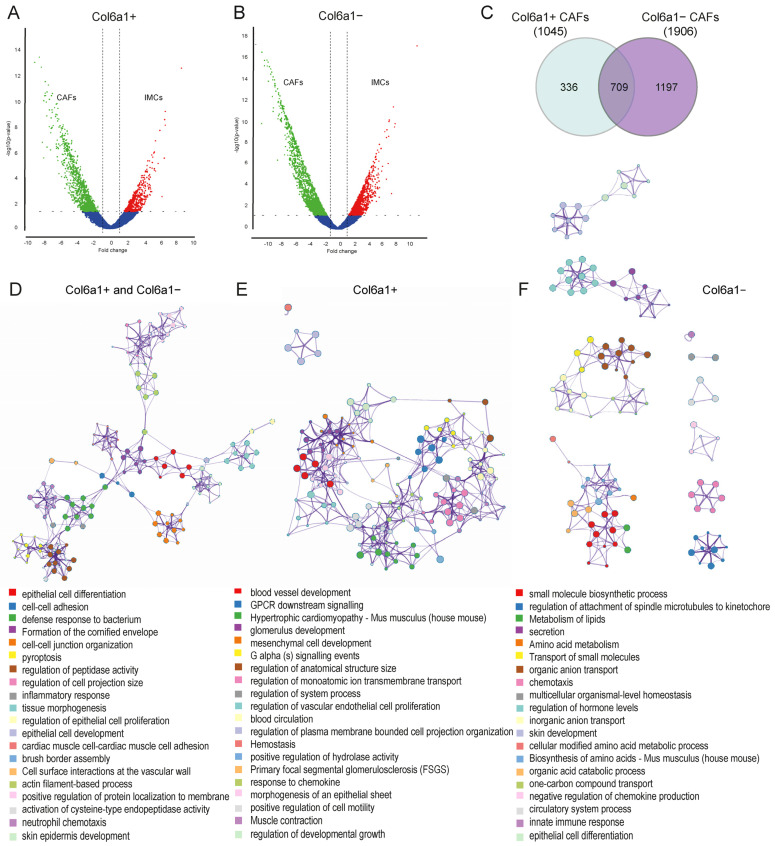
Col6a1+ and Col6a1− CAFs are activated upon AOM/DSS colon carcinogenesis. Volcano plots of deregulated genes in (**A**) Col6a1+ CAFs versus normal Col6a1+ IMCs and (**B**) Col6a1− CAFs versus normal IMCs. (**C**) Venn diagram showing the differential and common upregulated genes in Col6a1+ and Col6a1− CAFs. (**D**) Network of enriched terms in Col6a1+ CAF and Col6a1− CAF common upregulated gene signature. (**E**) Network of enriched terms in Col6a1+ CAF unique gene signature. (**F**) Network of enriched terms in Col6a1− CAF unique upregulated gene signature. Networks are colored by cluster ID, where nodes that share the same cluster ID are typically close to each other (generated through metascape.org).

**Figure 3 ijms-25-00148-f003:**
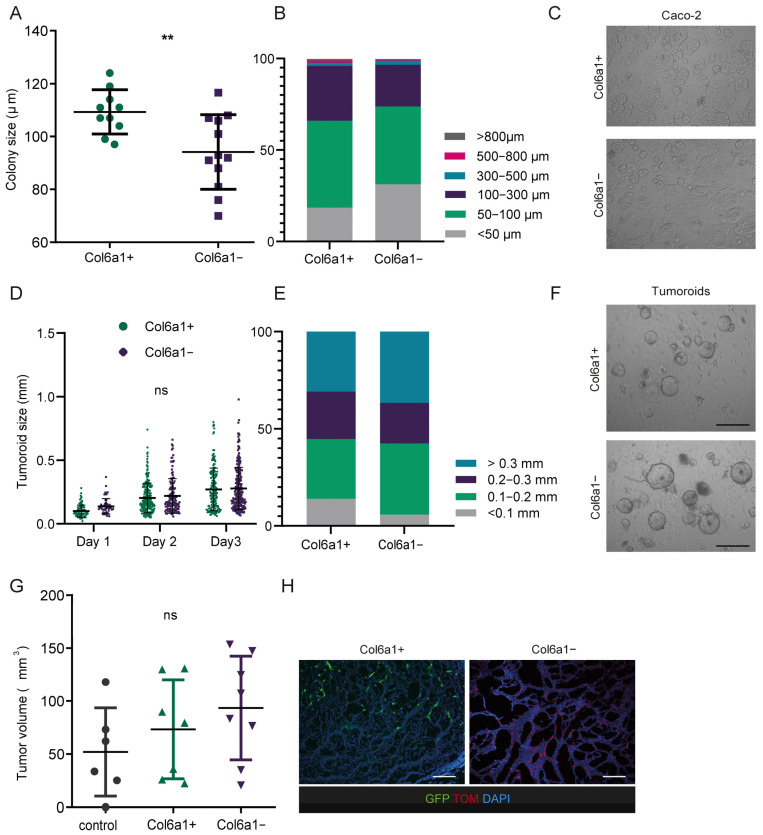
Co-culture experiments and allografts show similar effects of Col6a1+ and Col6a1− CAFs on cancer cell growth. (**A**) Average colony size per well, (**B**) size distribution, and (**C**) representative bright field images of Caco-2 colonies after 3 days of culture on sorted Col6a1+ and Col6a1− colonic IMCs. Data represents mean ± SD from one of three experiments performed, n = 10–12 wells, ** *p* < 0.01, Scale bar = 0.5 mm. (**D**) Tumoroid size per condition for each day of the co-culture, (**E**) size distribution of tumoroids at day 3 of the co-culture, and (**F**) representative bright-field images of AOM/DSS tumoroids at day three of their co-culture with sorted Col6a1+ and Col6a1− colonic IMCs. Data represents mean ± SD of tumoroids from one of three experiments performed. ns = not statistically significant, Scale bar = 1 mm. (**G**) Total volume of allografts after 15 days of growth with Col6a1+ and Col6a1− colonic IMCs. Data represents mean ± SD from one of three experiments performed (n = 6–8), ns = not statistically significant. (**H**) Representative fluorescent images of allografts with sorted Col6a1+ and Col6a1− colonic IMCs. Scale bar = 50 μm.

**Figure 4 ijms-25-00148-f004:**
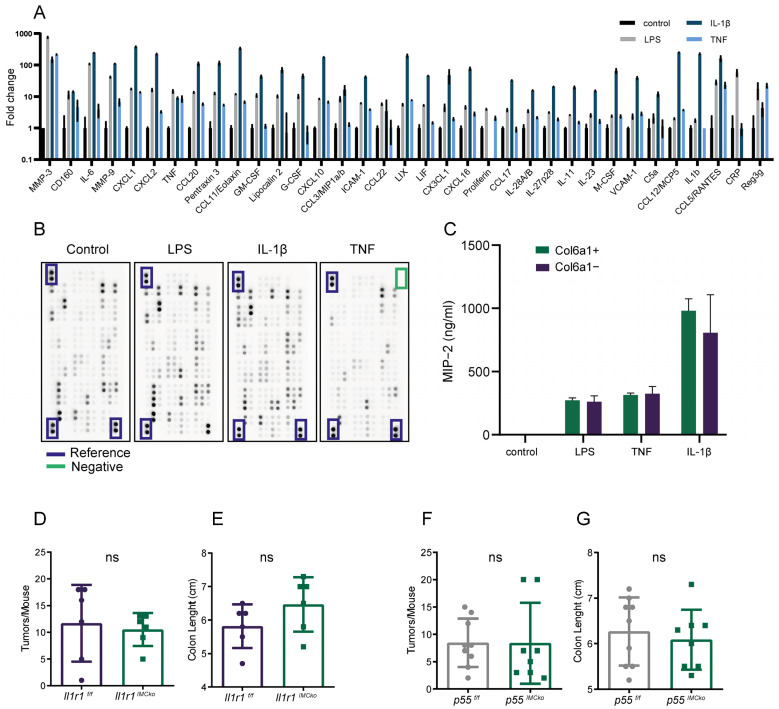
Deletion of IL-1R1 and TNFR1 in Col6a1+ IMCs is not sufficient to ameliorate CAC. (**A**) Proteome profiling of cultured unsorted IMCs upon LPS, IL-1β, and TNF stimulation. Only factors with differences in fold change > 2 in at least one condition are shown. Data represents mean ± SD from one experiment performed in duplicates. (**B**) Image showing the signal intensity of the proteome profile assay as obtained from the ChemiDoc XRS^+^ instrument. (**C**) MIP-2 quantification in the supernatants of Col6a1+ and Col6a1− IMCs stimulated for 24 h with LPS, TNF, and IL-1β. One representative of two independent experiments performed in triplicates is presented. (**D**) Number of tumors per mouse and (**E**) colon length in *Il1r1*^IMCko^ mice (n = 6) and their littermate controls (n = 6) at the end of the AOM/DSS protocol (one representative experiment of four performed). (**F**) Number of tumors per mouse and (**G**) colon length in *p55*^IMCko^ mice (n = 8) and their littermate controls (n = 9) at the end of the AOM/DSS protocol (one representative experiment of two performed). ns, not statistically significant.

**Table 1 ijms-25-00148-t001:** Antibodies used in flow cytometry and immunohistochemistry.

Antigen	Conjugate	Clone/Cat. Number	Company	Use	
CD45	A700	30-F11	BioLegend, San Diego, CA, USA.	FC	Lineage-Antibodies
CD326 (EpCAM)	APC-efluor780	G8.8	eBioscience, Waltham, MA, USA.	FC	
Ter119	APC-efluor780	TER-119	eBioscience, Waltham, MA, USA.	FC	
CD31	APC/Fire 750	390	BioLegend, San Diego, CA, USA.	FC	
CD201	PE/Cy5	eBio1560	Invitrogen, Waltham, MA, USA	FC	
α-SMA	unconjugated	1A4	Sigma, St. Louis, MO, USA	IHC	
Anti-mouse-IgG	A647	A-21235	Invitrogen, Waltham, MA, USA	IHC	Secondary

FC, Flow Cytometry; IHC, Immunohistochemistry.

## Data Availability

The RNA-seq datasets generated and analyzed during the current study are accessible through the GEO Series accession number GSE247089.

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
