# Peer review of "Activation and Functions of Col6a1+ Fibroblasts in Colitis-Associated Cancer"

_ijms, 2023, doi:10.3390/ijms25010148_

Round 1
Reviewer 1 Report
Comments and Suggestions for Authors
In this work Chalkidi and colleagues analysed the properties of cancer-associated fibroblasts (CAFs) that show expression of collagen 6a1 using colitis associated CRC models in genetically modified mice. The manuscript is nicely written and builds further on their previously published work with the mouse models and adds to the current knowledge. Although the ultimate figure does not show a significant difference between groups it is important to share these data with the scientific community. Below I listed some comments to improve the manuscript
Major comments
1. To what extent were the lesions observed in the AOM-DSS model actual cancers? In many studies this model is described to generate advanced adenomas with high grade dysplasia, but because of their abundant number/growth and humane endpoints invasive cancers are not so often reported. This is important in the light of the conclusions drawn. An example of histology showing invasion would be nice.
2. Interesting to note that the AOM-DSS model is a neutrophil dependent model, so the association with neutrophil recruitment is not surprising, but the fact that CAFs might mediate this is very interesting. Is there a relation between the Col6a1 subset and neutrophil recruitment? Did/could the authors investigate with some in vitro neutrophil attraction assays?
3. Figure 3, colony formation is more associated with a soft agar assay, while the data reported here are basically looking just at proliferation. In figure 3D are these cocultures in the sense of direct contact cocultures? Is there any change on the cellular behaviour/organisation? Previously it has been show that different subset or normal versus cancer fibroblasts can change the cellular organisation considerably.
4. Why did the authors engraft a coculture of Col6a1+ and Col6a1- cells with MC38 and not with the AOM/DSS tumeroids, which would better reflect the situation? Did the authors check immune infiltration and total stromal accumulation in these mouse models? This would be very valuable information to add.
5. In figure 4A/B are these unsorted fibroblasts? Meaning both Col6a1+ and Col6a1-? Line 231 I do not understand how the authors end up at this conclusion.
6. Some histology on the experiments performed in Fig4 would be nice/informative to assess the histological changes//immune infiltration/stromal accumulation.
Minor comments
1. Please check statistics throughout, in figure 3 the data should represent a quadruplicate, but more datapoints are presented in the figure.
Reviewer 2 Report
Comments and Suggestions for Authors
The article presented by Niki Chalkidi and collaborates, entitled “Activation and functions of Col6a1+ fibroblasts in colitis-associated cancer”, is an original article whose objective is to analyse the properties of Col6a1+ cancer associated fibroblasts (CAFs) in intestinal carcinogenesis and define their pathophysiological significance, focusing on their innate immune activation. the properties of Col6a1 positive fibroblasts in colitis associated cancer. The authors have published different articles with Col6a1Cre mica. The article is interesting, it uses numerous invivo models which praises the work done . However, the writing of the article has important deficits in the explanation of the experimental groups which makes it difficult to follow the article correctly.
Major revision:
1. In results. Line 94 the authors do not explain the reason for isolating the Col6a1-TOM+ population, nor do they even indicate that it is TOM and the reason for its choice. Nor is it explained in the introduction.
2. The same with the CD201 marker, the authors must explain their choice (Line 99)
3. The references of the articles that use these markers of the telocytes (line 96) and the tropocytes (line 101) must be attached.
4. The Line 69 “Further analysis indeed showed that the Col6a1Cre strain selectively targets a distinct fraction of IMCs in homeostasis and CAFs in intestinal cancer [21]”. The information should be expanded since then the authors use those cellular groups as a comparison group and there is not enough information in the introduction to make you understand their results
5. Explain the reason of injected MC38 cells. Line197
Minor revision:
1. Line 47 acronym: PDGFR
2. Line 55 acronyms: TGFβ and IL-1β
3. Line 64 acronyms: IKK2, and NFκB
4. Line 69 acronym: IMCs
5. Line 77 acronym: AOM/DSS
6. Figure 3. Scale
Round 2
Reviewer 1 Report
Comments and Suggestions for Authors
My comments have been addressed
Reviewer 2 Report
Comments and Suggestions for Authors
The authors have introduced the suggested changes